# No changes in overall AMOC strength in interglacial PMIP4 timeslices

Zhiyi Jiang[1], Chris Brierley[1], David Thornalley[1], and Sophie Sax[1]

[1]Department of Geography, University College London, London, WC1E 6BT, UK

**Correspondence:** Zhiyi Jiang (z.jiang.17@ucl.ac.uk)

**Abstract.** The Atlantic Meridional Overturning Circulation (AMOC) is a key mechanism of poleward heat transport and an important part of the global climate system. How it responded to past changes in forcing, such as experienced during Quaternary interglacials, is an intriguing and open question. Previous modelling studies suggest an enhanced AMOC in the mid-Holocene compared to the preindustrial period. In earlier simulations from the Palaeoclimate Modelling Intercomparison Project (PMIP), this arose from feedbacks between sea ice and AMOC changes, which were dependent on resolution. Here we present an initial analysis of the recently available PMIP4 simulations, which include three experiments during interglacials – the previous one at 127,000 years ago (127 ka BP, called *lig127k*), one in the middle of the Holocene (*midHolocene*, 6 ka BP) and a preindustrial control simulation (*piControl*, 1850 CE). Both *lig127k* and *midHolocene* have altered orbital configurations compared to the *piControl*. The ensemble mean of the PMIP4 models shows the strength of the AMOC does not markedly change between the *midHolocene* and *piControl* experiments or between the *lig127k* and *piControl* experiments. Therefore, it appears orbital forcing itself does not alter the overall AMOC. We further investigate the coherency of the forced response in AMOC across the two interglacials, along with the strength of the signal, using eight PMIP4 models which performed both interglacial experiments. Only 2 models show a stronger change with the stronger forcing, but those models disagree on the direction of the change. We propose that the strong signals in these 2 models are caused by a combination of forcing and the internal variability. After investigating the AMOC changes in the interglacials, we further explored the impact of AMOC on the climate system, especially on the changes in the simulated surface temperature and precipitation. After identifying the AMOC's fingerprint on the surface temperature and rainfall, we demonstrate that only a small percentage of the simulated surface climate changes can be attributed to the AMOC. Proxy records of sedimentary Pa/Th ratio during the two interglacial periods both show a similar AMOC strength compared to the preindustrial, which fits nicely with the simulated results. Although the overall AMOC strength shows minimal changes, future work is required to explore whether this occurs through compensating variations in the different components of AMOC (such as Iceland-Scotland overflow water). This line of evidence caution against interpreting reconstructions of past interglacial climate as being driven by AMOC, outside of abrupt events.

## 1 Introduction

The Atlantic Meridional Overturning Circulation (AMOC) is a large system of ocean currents involving differences of the temperature and salinity between the water in the tropics and North Atlantic Ocean (Rahmstorf, 2006). It consists of an upper

limb of warm and salty northward surface flow (the North Atlantic warm current, down to roughly 1200 m depth), and a lower limb of colder and deep southward flow (the North Atlantic Deep Water, 1500-4000 m depth) (Buckley and Marshall, 2016). The AMOC acts as a heat pump at the high latitudes as the meridional transportation brings warm water to the colder sub-polar and polar regions (Chen and Tung, 2018), then further modifies the climate in Northern Europe and the east coast of North America (Găinuşă-Bogdan et al., 2020). It is responsible for producing about half of the global ocean's deep waters, sourced from the northern North Atlantic (Petit et al., 2021).

Since the AMOC plays a vital role in air-sea interactions, along with its ability to transport and redistribute heat and its effect as a carbon sink in the Northern Hemisphere (Gruber et al., 2002), studying the evolution of the AMOC strength in the past is of great importance for us. It helps us identify the mechanisms which lead to the AMOC changes (Buckley and Marshall, 2016) and make projections for the future climate. Comparison of the AMOC changes between different geological eras can provide us with a better understanding of the roles of the external forcing in the AMOC strength variations. In addition, past AMOC variations suggest that the distribution of surface heat and freshwater flux can affect the location of deep water formation and result in transient changes in the AMOC (Kuhlbrodt et al., 2007).

Our study focuses on investigating the AMOC changes during the two interglacials – the Holocene (11.5 ka BP - 1950 CE) and Last Interglacial (130-115 ka BP). Two time slice experiments, the *midHolocene* (representing 6 ka BP) and the *lig127k* (representing 127 ka BP), have been selected by the Palaeoclimate Modelling Intercomparison Project Phase 4 (PMIP4) (Kageyama et al., 2018). 6 ka BP was chosen as warmest point during the Holocene thermal maximum according to existing surface temperature reconstructions (Joussaume and Braconnot, 1997), although this is being re-evaluated at present (Marsicek et al., 2018). The *midHolocene* is one of the entry cards (Kageyama et al., 2018) for the PMIP4 component of the current phase of the Coupled Model Intercomparison Project (CMIP6). The 127 ka BP is chosen as it represents the peak boreal warmth in the last interglacial (Capron et al., 2017a), and it has been identified as a period of high interest, due to its higher average global temperature and sea level (Capron et al., 2017b; Otto-Bliesner et al., 2017). It is considered a natural experiment for what climate may look like in the future and addresses one of CMIP's key questions: "How does the Earth system respond to forcing?" (Eyring et al., 2016). In the context of PMIP4, the focus lies on changes in insolation arising from the differences in Earth's orbit, while the greenhouse gases (GHG) concentrations were similar to that in the *piControl* experiment, and the continental configuration (ice-sheet distribution and elevations, land–sea mask, continental topography and oceanic bathymetry) were prescribed as the same as in *piControl* (Otto-Bliesner et al., 2017).

During the two interglacial periods, the orbital parameters are prescribed according to Berger and Loutre (1991). Eccentricity, the deviation of the Earth's orbit from a perfect circle, was larger (more elliptic) than that during the preindustrial period, especially for the *lig127k*. Meanwhile, perihelion, the closest point in the orbit to the sun, occurred much closer to the boreal summer solstice in the *lig127k*. Obliquity, the tilt of the Earth's axis, was also higher during these two warm periods (Otto-Bliesner et al., 2017). This leads to a positive Northern Hemisphere summer insolation anomaly at both 127 ka and 6 ka, compared to preindustrial, while the difference in annual incoming insolation at the top of the atmosphere between the two periods is marginal (see Fig. 3b of Otto-Bliesner et al., 2017, for the seasonal distribution of insolation). Due to the model differences in the internal model calendar and the impact of eccentricity and precession (the orientation of Earth's rotational

axis) on the length of the seasons, the date of the vernal equinox must be fixed in all simulations to 21st March (Joussaume and Braconnot, 1997; Otto-Bliesner et al., 2017). More detail on the forcings and boundary conditions for the *lig127k* and *midHolocene* experiment can be found in Otto-Bliesner et al. (2017) and in Eyring et al. (2016) for the *piControl* experiment. Based on the experimental setup, the *midHolocene* and *lig127k*, when the seasonal insolation is the strongest forcing, are two reasonable periods to study whether or not the changes in orbital forcing have altered the overall AMOC strength in the two past interglacials compare to the *piControl* experiment.

After introducing the methods used in this study (Sect. 2), we first analyse the behaviour of AMOC during the Quaternary interglacials in individual PMIP4 models in Sect. 3, then we explore the AMOC variations during the past two interglacials based on the models ensemble mean, which are shown in Sect. 3.1. In Sect. 3.2, as the seasonal changes in incoming solar radiation amplified in the *lig127k* compared to the *midHolocene*, we investigated further to see whether the simulated response show similar amplification in these individual models or not. Meanwhile, we also devised a series of tests that must be passed for a forced response, and also try to identify the causes for the changes in AMOC that we see in individual models. Furthermore, since we have identified that the AMOC changes could leave a fingerprint on the surface temperature and precipitation variation in the *midHolocene*, as well as in the *piControl*, regressions of surface conditions against AMOC have been computed for each simulation runs for both *midHolocene* and *piControl*, and they are shown in Sect. 4.1. Based on the computed AMOC's fingerprint on the surface temperature and precipitation in individual models, we further estimated the percentage of simulated surface temperature and precipitation changes that could be explained by AMOC changes in Sect. 4.2, which is generally shown as the AMOC's role in global surface climate changes. After investigating the changes in AMOC and the role of AMOC in climate system in PMIP4 simulations, comparisons with proxy reconstructions for the Holocene and the last interglacial are discussed in Sect. 5.

## 2   Methods

To be included in this study, a model must have performed an experiment following either the protocol for the *midHolocene* or *lig127k* as laid out by Otto-Bliesner et al. (2017), and then archived the output of this experiment onto the Earth System Grid Federation. Twelve CMIP6 models have provided the necessary output of zonal-mean ocean meridional overturning mass streamfunction (called 'msftmz' or 'msftmyz' depending on the grid used) to undertake our analysis. Of these, eight models performed both interglacial experiments. Details of the individual models are shown in Tab. 1.

The AMOC intensity is computed using a modified version of Climate Variability Diagnostic Package (Phillips et al., 2014; Danabasoglu et al., 2012). Rather than using principal component analysis to define the AMOC (Danabasoglu et al., 2012), the maximum overturning streamfunction at 30°N is used (Zhao et al., 2022). Patterns of surface climate association with AMOC variations were computed via linear regression with the AMOC timeseries; precipitation regressions have been encoded to complement the existing surface temperature patterns (Zhao et al., 2022).

The maximum AMOC strength is defined as the maximum of the annual mean meridional mass overturning streamfunction below 500 m at 30°N (and additionally at 50°N). The simulated maximum AMOC strength at these two latitudes from individ-

ual models are used for the comparisons of the changes in the maximum AMOC between the interglacials and preindustrial. The maximum AMOC usually occurs between 30°N to 40°N, yet the RAPID-MOCHA mooring array locates at 26°N, hence the 30°N is chosen. The choice of 50°N is due to the location of the OSNAP section (53-60°N). The data from the two arrays can provide us with estimations of the present-day AMOC strength (Rayner et al., 2011; Lozier et al., 2019) for reference (the observation period of RAPID mooring array started in 2004, and the OSNAP started in 2014). If the latitudes of, say, 35°N or 55°N had been selected instead, the impacts on the results are subtle (Brierley et al., 2020) and would not effect the conclusions presented here.

**Table 1.** Model simulation length (after spin-up, in years) and their AMOC at 30°N (in Sv, also with the standard deviation). The data from FGOALS-f3-L used for the preindustrial conditions comes from the *historical* simulation for years 1850 to 1899, as the AMOC variable is unavailable for the *piControl* simulation.

| Model | Reference | Preindustrial | | midHolocene | | lig127k | |
|---|---|---|---|---|---|---|---|
| | | Length | AMOC | Length | AMOC | Length | AMOC |
| CESM2 | Otto-Bliesner et al. (2020) | 500 | 19.1±0.8 | 700 | 19.4±0.8 | 700 | 19.9±0.7 |
| EC-Earth3-LR | Zhang et al. (2021) | 201 | 15.0±2.1 | 203 | 16.2±2.7 | 210 | 18.6±1.4 |
| FGOALS-f3-L | Zheng et al. (2020) | 50 | 23.9±2.7 | 200 | 24.4±2.2 | 500 | 25.2±2.1 |
| FGOALS-g3 | Zheng et al. (2020) | 699 | 32.8±2.5 | 500 | 33.5±1.9 | 500 | 33.4±2.1 |
| GISS-E2-1-G | Kelley et al. (2020) | 851 | 24.4±2 | 100 | 24.5±1.6 | 100 | 25.0±1.8 |
| HadGEM3-GC31-LL | Williams et al. (2020) | 100 | 17.0±1.2 | 100 | 18.4±1.2 | 100 | 18.1±1.1 |
| IPSL-CM6A-LR | Lurton et al. (2020) | 1200 | 12.1±1.3 | 550 | 11.6±1.3 | 550 | 10.3±1.3 |
| NorESM2-LM | Seland et al. (2020) | 391 | 21.2±0.9 | 100 | 21.4±0.8 | 100 | 21.6±0.8 |
| INM-CM4-8 | Volodin et al. (2018) | 531 | 17.1±1.3 | 200 | 16.3±1.1 | N/A | N/A |
| MPI-ESM1-2-LR | Scussolini et al. (2019) | 1000 | 20.1±1.2 | 500 | 20.1±1.4 | N/A | N/A |
| MRI-ESM2-0 | Yukimoto et al. (2019) | 701 | 18.0±1.0 | 200 | 20.2±1.4 | N/A | N/A |
| ACCESS-ESM1-5 | Yeung et al. (2021) | 900 | 19.5±1.1 | N/A | N/A | 200 | 22.5±1.6 |

All models are regridded on to a common 1° latitude grid with 61 levels of depth between 0-6000 m in the ocean to compute ensemble averages. All simulations are given equal weighting when the ensemble mean change in AMOC is computed.

A fingerprint of the AMOC on wider climate is computed separately for each simulation. The fingerprints are obtained by linearly regressing temperature / precipitation at each grid box over the globe onto AMOC maximum at 30°N, using the equation: $\delta T = \alpha \delta \Psi_{30N} + c$, where $\delta$ indicates an anomaly within a simulation, $T$ is the temperature (at the grid point), $\Psi$ the maximum overturning streamfunction at 30°N in the Atlantic, $\alpha$ is the fingerprint coefficient and $c$ is a constant. A 15-month low-pass Lanczos filter is applied to the AMOC timeseries prior to computing the regression. Precipitation fingerprints are

computed using percentage variations, rather than absolute rainfall anomalies. The percentage of local surface temperature changes that can be explained by AMOC changes, is then estimated by comparing simulated changes to the AMOC change multiplied by the regression coefficient (averaged between the interglacial and preindustrial simulations) ($\Delta T_{\Psi}/\Delta T$). Similarly, the percentage of local precipitation changes that can be explained by the AMOC changes in each simulation can also be computed. However, in order to provide a less messy figure, the ensemble mean plot of the percentage of precipitation changes that can be explained by AMOC changes has been made instead. Firstly, we regridded all the models onto a common $1° \times 1°$ grid, then compute the ensemble mean AMOC-induced precipitation changes ($\overline{\Delta P_{\Psi}}$), and the ensemble mean simulated precipitation changes ($\overline{\Delta P}$). Eventually, the ratio ($\overline{\Delta P_{\Psi}}$ / $\overline{\Delta P}$) provides us with the final results. Directly taking the averages based on each model's ratio is not used, as it leads to a chaotic image due to division by minimal $\Delta P$.

## 3   Simulated AMOC during the *midHolocene* and *lig127k*

The first-order determinant on the AMOC strength is the model used for the simulation (Tab. 1). Fig. 1a clearly shows that the *piControl* AMOC strength at 30°N is highest in FGOALS-g3, with the rest of the models generally have results ranging from about 12-24.5 Sv. The highest simulated AMOC strength is more than twice of the lowest ones, even without FGOALS-g3. The dashed green and pink horizontal lines in Fig. 1a provide us with the information from direct observations in recent decades. The observational AMOC strength at 26°N which comes from the RAPID array (Rayner et al., 2011) is stronger than that at 53-60°N from the OSNAP section (Lozier et al., 2019; Srokosz et al., 2012). The AMOC strength from the simulation runs also reveals that the AMOC is stronger in the sub-tropical region than that in the sub-polar region (except for ACCESS-ESM1-5 and FGOALS-g3, Fig. 1a). Furthermore, in the INM-CM4-8 model, the *piControl* AMOC strength at 30°N is 11 Sv stronger than that at 50°N. The much stronger AMOC at 30°N compared to that at 50°N also occurs the GISS-E2-1-G and NorESM2-LM models in the *piControl* experiment, they demonstrate differences of 9 Sv and 7.6 Sv, respectively. Differences in other models generally are between 2 and 4 Sv. It should be noted that the observed AMOC strength at 53-60°N is computed in density coordinates (Lozier et al., 2019; Srokosz et al., 2012), whilst all the other values are computed in depth coordinates.

The large spread in the simulated AMOC strength seen in the *piControl* experiment raises questions about whether the models can accurate simulate changes in AMOC (Eyring et al., 2021). The spread is an unfortunate feature of both the wider CMIP6 and CMIP5 ensembles (e.g. Xie et al., 2022). Disappointingly the uncertainty in modern-day oceanographic observations is such that (17.0±4.4 Sv measured by the RAPID array, Frajka-Williams et al., 2019) few of the simulations can be categorically ruled out (Weijer et al., 2020). The *piControl* experiment represents an earlier time than that of the observations, and AMOC is known to have changed between them (Thornalley et al., 2018; Caesar et al., 2018). However, the changes in AMOC seen the historical simulations (Gong et al., 2022) are relatively small compared to the differences between the models and observations, meaning that the consequences of the temporal offset are not important compared to model biases.

The extremely low value in the IPSL-CM6A-LR model is mainly caused by the inaccurate representation for the overflow waters or deep western boundary current (DWBC) and biases in the precipitation in the Norththe *piControl* experiment represents an earlier time than that of the observations, the changes in AMOC over the historical simulations (Gong et al., 2022) are

insufficient to account for any differences Atlantic Ocean, which are challenging to resolve in climate models (Boucher et al., 2020). In some models, the lack of overflow parameterizations which commonly occur in low resolution models (Danabasoglu et al., 2014) could be another reason for the slightly underestimated simulated AMOC strength. The AMOC strength in the FGOALS-g3 model at both sub-tropical and sub-polar regions is very high, even compared to its predecessor FGOALS-g2 (Li et al., 2020). These can be attributed to the strong convection occurring in the North Atlantic Ocean. Despite a mixed layer depth similar to observations, the intensity of the simulated deep convection is too strong in the Irminger, Labrador and Nordic Seas, and wintertime convection was overestimated (Li et al., 2013).

The interannual variability of the AMOC is also model-dependent (Tab. 1), and generally does not alter much between the various experiments. EC-Earth3-LR and the two models by FGOALS are the exceptions, but they do not provide coherent message about the response to increasing orbital forcing. Therefore, we consider these to be different samples from the same underlying distribution (Latif et al., 2022).

The absolute AMOC changes in the *midHolocene* and *lig127k* experiments (with respect to the *piControl*) are compared to the magnitude of the internal variability (1 standard deviation) of each model's *piControl* experiment (Fig. 1b). The magnitude of the simulated AMOC changes in the *midHolocene* are within the range of the model's internal variability for all the models, except for HadGEM-GC31-LL and MRI-ESM2-0. The extent of AMOC changes in the *lig127k* are generally larger, with 3 models (ACCESS-ESM1-5, EC-Earth3-LR and IPSL-CM6A-LR) showing changes that are larger than their internal variability.

Looking at the relative changes in AMOC seen in the *midHolocene* and *lig127k* experiments (Fig. 1c) is one possible way to approach to account the large spread in AMOC strengths in the *piControl*. Changes within the $\pm 5\%$ range (red dashed horizontal lines) have previously been considered to represent no substantial AMOC changes (Brierley et al., 2020). Only 3 out of 11 PMIP4 models which performed *midHolocene* experiment demonstrate a change in *midHolocene* AMOC larger than this. In addition, these 3 models all show a stronger AMOC in the *midHolocene* than that in the *piControl* experiment. The majority of the models do not demonstrate a substantial change in the maximum AMOC strength between this two timeslices either at 30°N or at 50°N (Tab. 1; Fig. 1c. Percentage of AMOC changes at 50°N is not shown). Brierley et al. (2020) state that these findings are consistent with the palaeo-reconstructions for the mid-Holocene, something discussed further in Sect. 5. Similar results are seen for the *lig127k* in Fig. 1c. There are 4 models (ACCESS-ESM1-5, EC-Earth3-LR, FGOALS-f3-L and IPSL-CM6A-LR) that have AMOC strength changes exceeding 5% of the *piControl* strength at both 30 and 50°N. (The changes in AMOC strength in HadGEM3-GC31-LL at 30°N is outside the 5% range as well, but its AMOC change at 50°N is not). The extent of these deviations are generally larger than those seen in the *midHolocene*.

### 3.1 Ensemble mean AMOC changes

To explore the spatial patterns of changes in the AMOC structure in past warm interglacials, we compute PMIP4 ensemble mean AMOC changes (Fig. 2). The overlaid contours display the model averaged AMOC pattern in the *piControl* experiment to help place these changes in context. The two plots do not reveal a substantial change in the AMOC strength at the location where the maximum AMOC occurs (35-40°N,1000 m). There is a slight increase about 0.4-0.8 Sv in the maximum AMOC strength in the *midHolocene*, growing to 1.0-1.4 Sv in the *lig127k*. There is a slight intensification of the *midHolocene* model

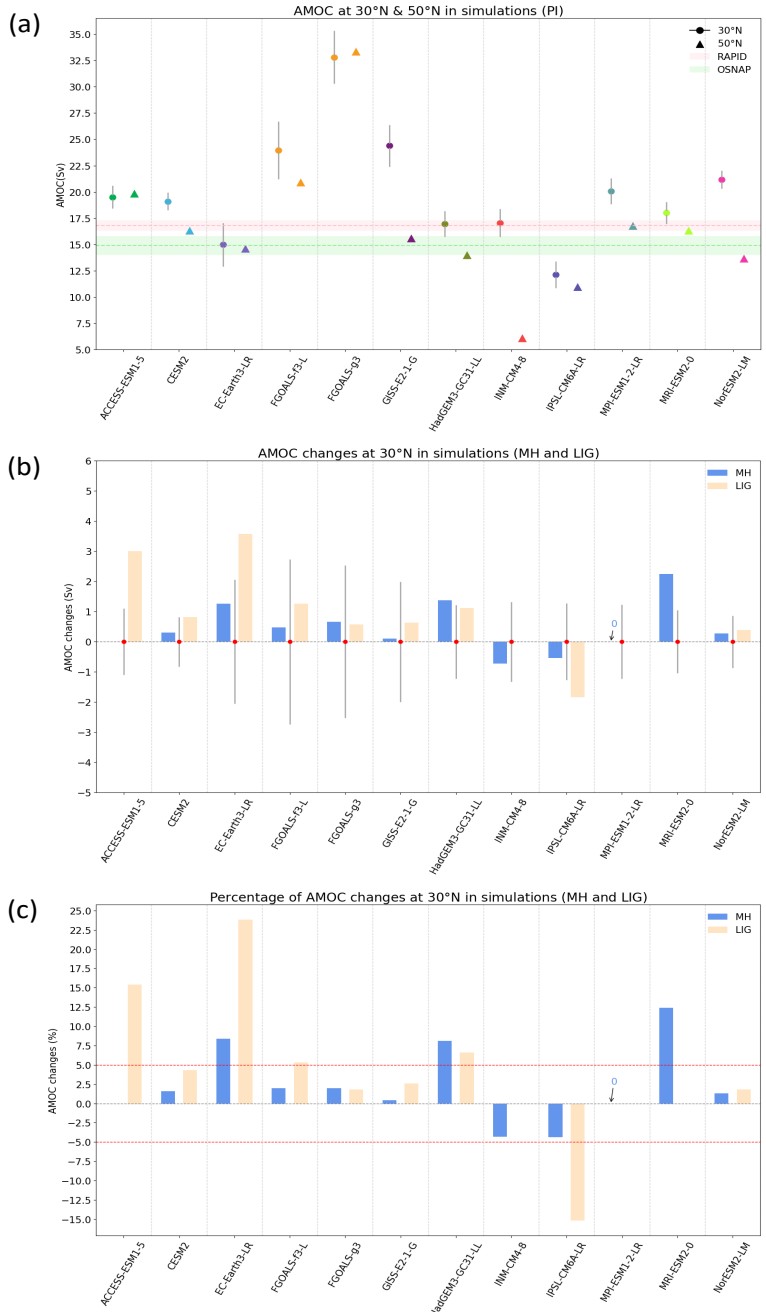

**Figure 1.** Maximum AMOC strength and AMOC changes. (a) Maximum AMOC strength at 30°N (circle legend) (with error bars which indicate 1 standard deviation) and 50°N (triangle legend) in preindustrial control simulations. Observational estimates of the present-day AMOC strength are shown from both the RAPID-MOCHA array (26°N) and the OSNAP section (53–60°N). (b) Absolute AMOC changes at 30°N in the *midHolocene* and *lig127k* experiments (w.r.t *piControl* AMOC). The error bars between the two histograms of each model show the magnitude of the internal variability (1 standard deviation) of each model's *piControl* experiment. (c) Percentage of AMOC changes at 30°N in the *midHolocene* and *lig127k* experiments (w.r.t *piControl* AMOC). Data within the ±5% range indicate no obvious AMOC changes. The number 0 is annotated in (b) and (c) as the MPI-ESM1-2-LR model does not shown any AMOC changes between the *midHolocene* and *piControl* (see Tab. 1).

averaged AMOC strength at depth (below 2000m, with the largest changes up to 1.0-1.2 Sv at 2500 m in the sub-tropics). The *lig127k* experiments do not show such a focus of their intensification at depth, with the largest changes occurring in the top 500 m (Fig. 2b). An overall stronger AMOC in the *lig127k* is confined at the low-mid latitudes, as the AMOC strength becomes weaker in the sub-polar and polar regions (north of 55°N). Since the *midHolocene* and *lig127k* ensembles contain some different models, we additionally analyse the pattern of AMOC changes between the *midHolocene* and *lig127k* only using

the models which have AMOC data in both of the periods (8 models in total). This demonstrates that the different increases in shallow (top 1000 m at low-mid latitudes) (*lig127k*, Fig. 2b) and deep (below 2000 m at 0-60°N) (*midHolocene*, Fig. 2a) branches are not an artefact of the additional models (Fig. 2c).

In all, although slightly larger changes in maximum AMOC are seen in *lig127k* than that in *midHolocene*, the maximum AMOC changes based on the ensemble mean during the past interglacials never exceed 1.5 Sv. This is definitely less than

185 10% of the respective *piControl* maximum strength, and generally less than 5%. There are some regions (such as at depth in the *midHolocene*) that show greater proportional signals. However as with the AMOC strength, there are differences in the intensities of the AMOC pattern between individual models, but considering creating ensembles means of the percentage changes instead does not robustly alter our conclusions (not shown).

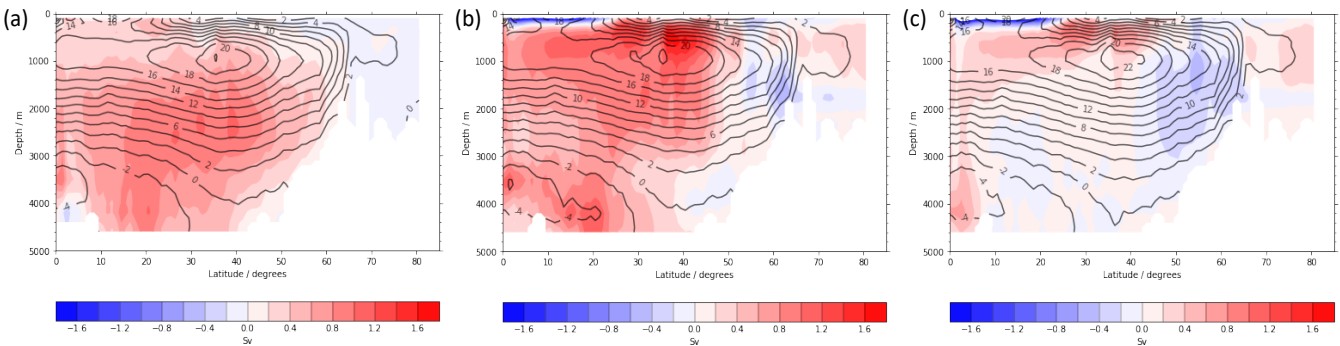

**Figure 2.** Ensemble, annual mean AMOC spatial structure changes in PMIP4. (a) Ensemble mean AMOC changes between the 11 PMIP4 models that have performed the *midHolocene* and *piControl* experiments. (b) Ensemble mean AMOC changes between the *lig127k* and *piControl* (consisting of 9 models). (c) Ensemble mean AMOC changes between the *lig127k* and *midHolocene* experiments (8 models). Overlaid black contours show model-averaged AMOC strength in the respective *piControl* simulations in (a) and (b), and in the respective *midHolocene* simulations in (c).

## 3.2 Assessing the forced response in AMOC

Since the seasonal changes in incoming solar radiation were amplified in *lig127k* compared to *midHolocene*, it would be expected (Williams et al., 2020) that the AMOC changes seen in the *lig127k* experiment are a similar, but stronger version of those seen in the *midHolocene* experiment. This is explored by analysing the AMOC profiles at 30°N for the 8 models which performed both interglacial experiments (Fig. 3). Only five models (CESM2, EC-Earth3-LR, FGOALS-f3-L, HadGEM3-GC31-LL and IPSL-CM6A-LR) show changes in AMOC in both experiments (at around 1000 m depth). The magnitude of ampli-

fication is very subtle in the CESM2 and FGOALS-f3-L models. The increases in AMOC shown between the *midHolocene* and *piControl* in the HadGEM3-GC31-LR model are actually slightly larger than those seen in the *lig127k* and attributed by Williams et al. (2020) as being a consequence of internal variability. The IPSL-CM6A-LR and EC-Earth3-LR model are the only 2, out of the 8 models, that demonstrate a noticeable, progressive changes from the *piControl* to *midHolocene* to *lig127k*. However, those 2 models show changes in opposite directions, with EC-Earth3-LR shows a positive response to the increased forcing, while the IPSL-CM6A-LR reveals a negative response.

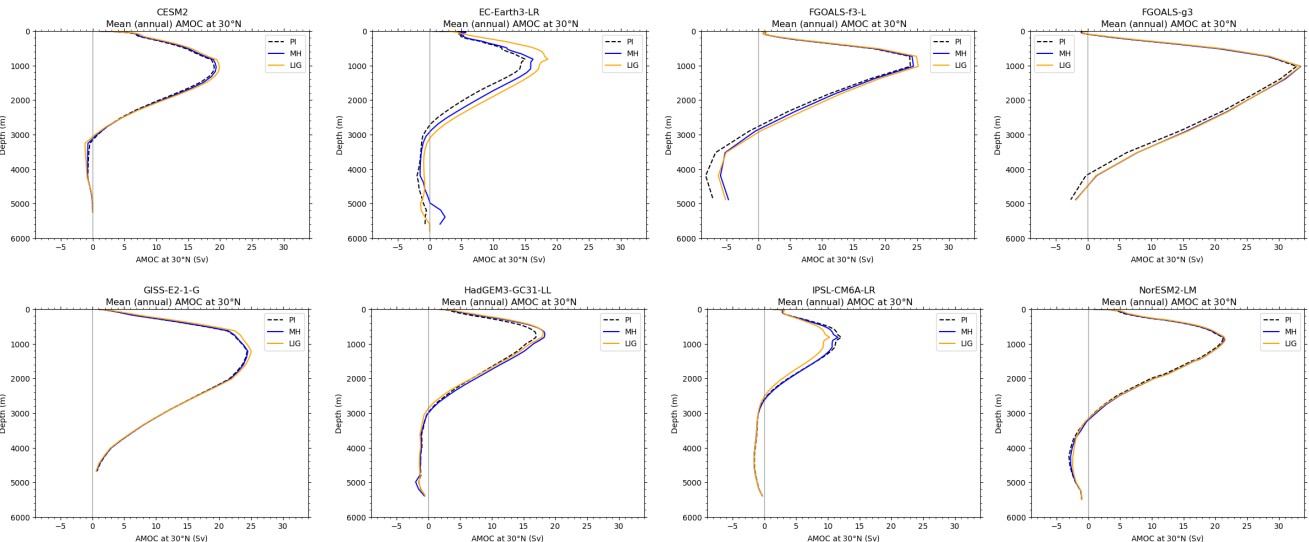

**Figure 3.** Mean (annual) AMOC profile at 30°N in simulations. The blue line shows the AMOC profile in the *midHolocene*, the amber line shows AMOC profile in the *lig127k*, and the dashed black line indicates *piControl* AMOC profile.

To demonstrate that AMOC responds to orbital forcing, one would look for the ensemble to simulate AMOC changes that are (i) extant, (ii) related to the strength of the forcing, (iii) detectable over the internal variability and (iv) model independent. Building on these criteria, we devise a series of tests that must be passed to show a forced response within a single experiment. Firstly, we test whether there is a change in AMOC, which here we arbitrarily take to be greater than 1 Sv. The changes in orbital configuration result in seasonal insolation shifts at the Northern high latitudes that are generally more than twice in the *lig127k* than in the *midHolocene* (Otto-Bliesner et al., 2017). The AMOC response may not be linear, so the second test sets a weaker threshold and looks at whether the AMOC changes in the *lig127k* are larger by half again than the *midHolocene* ones.

Assessing whether any AMOC changes are detectable against a model's internal variability in its AMOC timeseries is challenging given the relative lengths of the simulations (Tab. 1) and the known existence of low-frequency variability in AMOC (e.g. Fischer, 2011; Shi and Lohmann, 2016; McKay et al., 2018; Bonnet et al., 2021). The relative role of internal variability is assessed by comparing its strength to the size of the changes in the long-term mean. Whether an individual model has substantial low frequency internal variability are evaluated by firstly applying a 25-year low pass Lanczos filter to the annual mean AMOC timeseries of each simulation, then the Pearson correlation coefficients (r) are computed between the

non-filtered timeseries and the filtered timeseries in each individual simulations. If the $r^2$ value is greater than 0.5, it suggests the low frequency variability dominates the AMOC timeseries (as it explains >50% of the AMOC variations). After inspection of the results for each individual simulation, we conclude that except for the IPSL-CM6A-LR and GISS-E2-1-G models, other models do not contain substantial low frequency variability according to this criteria. The IPSL-CM6A-LR is the only model for which all 3 experiments demonstrate substantial low frequency variability (Tab. 2). However, despite the CESM2 and EC-Earth3-LR models not meeting our particular criteria, the standard deviations of the filtered AMOC timeseries in these 2 models are at least half or more of the standard deviation of the non-filtered timeseries. Therefore, this suggests that low frequency variability plays an important role the CESM2 and EC-Earth3-LR models, even if it does not dominate the variability.

Only one of the eight models, EC-Earth3-LR, shows changes in AMOC that are categorised as both (i) extant and (ii) related to the strength of the forcing (Tab. 2). However, it is unclear if even these changes pass the 3rd criteria of detectability above internal variability - the amplitude of the *midHolocene* changes are less than one standard deviation of the interannual timeseries and there is also a confirmed presence of low frequency variability in the EC-Earth3-LR simulation (Zhang et al., 2021).

Clearly, the results of the individual tests performed here will depend somewhat on the criteria chosen. For example, if it is the maximum AMOC across all latitudes (rather than at 30°N), then both CESM2 experiments would show extant AMOC changes (Otto-Bliesner et al., 2020), but then signal ratio is only 1.3 rather than the 2.7 in Tab. 2. However, two conclusions will remain robust to the many possible permutations. Overall the ensemble does not show a consistent AMOC signal from the imposed forcing changes. In fact, not a single one of the eight PMIP4 models that have performed both the *midHolocene* and *lig127k* experiments show changes in AMOC strength that are unambiguously a response to the orbital forcing.

## 4 AMOC and global surface climate changes

### 4.1 Unchanging AMOC fingerprints

We further investigate the role of AMOC in the interglacial climate system, particularly looking at the impact of AMOC on the simulated surface temperature and precipitation changes. First, we regress the temperature and precipitation at each grid box over the globe onto AMOC maximum at 30°N for each simulation to obtain the local response to a 1 Sverdrup increase (see Sect. 2). Larger regression coefficients indicate that the interdecadal variability in the AMOC has more impact on the surface temperature or precipitation changes at each grid box. There is a strong relationship between AMOC change and surface temperatures over the northern North Atlantic, and they are most obvious in the Nordic Seas, south of Greenland, Labrador Sea and along the track of the Gulf stream (Fig. 4). This reveals that the AMOC has a noticeable influence on modulating the surface temperature through heat transport in those regions (Borchert et al., 2018; Jungclaus et al., 2014). The regression coefficients are generally higher in the Nordic Seas than that in the area in south of Greenland when referring to all the 11 PMIP4 models involved (not shown). The area of influence is generally confined to the northern North Atlantic (Fig. 4), although the FGOALS-g3-L and GISS-E2-1-G models both have particularly low coefficient values ($\sim 0.15$) even there

**Table 2.** Tests for assessing an orbitally forced response within a model. The first 2 tests are based on the AMOC changes in the *midHolocene* and *lig127k* compared to *piControl*, where the change greater than 1 Sv is highlighted. The third test is based on the ratio of the AMOC changes in the *lig127k* to the AMOC changes in the *midHolocene*, and it is highlighted when the signal ratio is greater than 1.5. The last 2 tests involve the internal variability. The standard deviation of the unfiltered and 25-yr low-pass filtered AMOC timeseries are computed by averaging the standard deviation for each model in all 3 experiments, weighted by the respective lengths (Tab. 1). The last row shows the number of experiments that have substantial low frequency variability ($r^2 > 0.5$) in each model based on the correlation between the non-filtered timeseries and 25-yr low-pass filtered timeseries. The r-value of all models in all 3 experiments are statistically significant ($p<0.05$), with the exception of FGOALS-f3 *piControl* experiment. It is possibly due to the short run length of just 50 years, as we use the historical experiment in this model to substitute the *piControl* experiment.

| Tests | CESM2 | EC-Earth3-LR | FGOALS-f3-L | FGOALS-g3 | GISS-E2-1-G | HadGEM3-GC31-LL | IPSL-CM6A-LR | NorESM2-LM |
|---|---|---|---|---|---|---|---|---|
| Δ *midHolocene* | *0.3* | **1.3** | *0.5* | *0.7* | *0.1* | **1.4** | *-0.5* | *0.3* |
| Δ *lig127k* | *0.8* | **3.6** | **1.3** | *0.6* | *0.6* | **1.1** | **-1.8** | *0.4* |
| Signal Ratio | **2.7** | **2.8** | **2.6** | *0.9* | *6.2* | *0.8* | **3.4** | *1.4* |
| Std Dev. (unfiltered) | 0.8 | *2.1* | *2.2* | *2.2* | *1.9* | *1.2* | *1.3* | 0.9 |
| Std Dev. (low pass filtered) | 0.4 | *1.1* | *0.6* | *0.8* | *0.9* | *0.5* | *0.9* | 0.4 |
| Sims w. Low freq. variability? | 0 | 0 | 0 | 0 | 1 | 0 | 3 | 0 |

(not shown). Here we present regression coefficients from the *midHolocene* simulations, yet these are effectively unchanged in either the *piControl* or *lig127k* simulations (not shown).

The AMOC temperature fingerprints in the North Atlantic are accompanied by a dipole response in precipitation (Fig. 4) with roughly a 5% decrease in the mid-latitude (30-50°N) and a 5% increase in the subpolar and polar regions. The largest AMOC-induced precipitation changes occur in the Tropics - with a reduction of about 10-15% in the Equatorial Pacific. The FGOALS-f3-L (not shown) and NorESM2-LM show a larger decrease than other models (20-30% and 30-40%, respectively). Low latitude (0-30°N) North Atlantic ocean generally reveals an increases (up to 10%) in rainfall as the AMOC changed by 1 Sv, and it is more obvious in IPSL-CM6A-LR and NorESM2-LM. The 25% / Sv change in the increasing of precipitation in NorESM2-LM, which can be explained by the northward shifting of the Intertropical Convergence Zone (ITCZ) due to the stronger AMOC at this region, and it further results in more precipitation. NorESM2-LM shows the largest changes across the whole globe (Fig. 4) and is somewhat of an outlier. The fingerprints are very similar if computed using either *piControl* or *lig127k* simulations (not shown) - demonstrating that influence of AMOC is robust feature in the models with minimal state dependence. It should be noted that despite these fingerprints being computed from analysis of the internal variability within individual simulations, the spatial patterns are clearly reminiscent of those seen in hosing experiments (e.g. Jackson and Wood,

2020). This demonstrates it is valid to assume that the teleconnection patterns associated with internally-generated changes in AMOC are the same as those from externally forced changes.

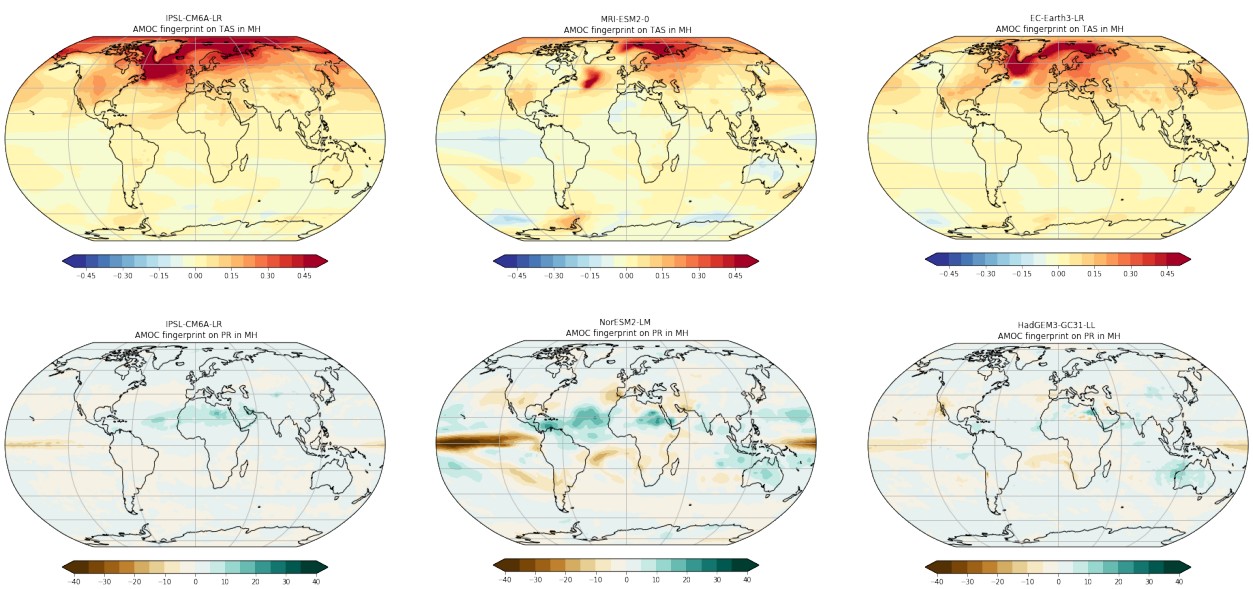

**Figure 4.** AMOC's fingerprint on the surface temperature (upper panels, unit: °C / Sv), and on the precipitation (lower panels, unit: % / Sv) in the *midHolocene* in selected PMIP4 models.

## 4.2 The role of AMOC in global surface climate changes between *midHolocene*, *lig127k* and PI

It is not uncommon to interpret terrestrial proxy records as being related to AMOC changes (e.g. Ayache et al., 2018), or to use compilations of proxy records to directly infer past AMOC changes (e.g. Ayache et al., 2018; Thornalley et al., 2018). Since

both the AMOC fingerprints and the changes in AMOC strength have been computed, we can determine maximum percentage of the local *midHolocene* climate changes that could be potentially be explained by the AMOC. Such analysis would help to identify regions where future proxy-based studies could be expected to contain an AMOC signal during the mid-Holocene.

Fig. 5 shows the percentage of the simulated surface air temperature changes that could potentially be explained by the AMOC changes in the *midHolocene*. The AMOC is only one of many factors influencing the local temperature changes.

For example, areas with percentage smaller than 0 can occur when the AMOC fingerprint suggests changes of the opposite sense as the actual changes. This percentage of fingerprint-estimated changes can approach, or even exceed, 100% when the *midHolocene* temperatures change is very small as simulated by the model when considering all factors. Both cases indicate that the AMOC changes can not explain the *midHolocene* temperature response in those areas.

The four models with the largest changes in maximum AMOC strength at 30°N are shown in the first 4 panels in Fig.

5. In general, those places that AMOC could explain half or more of the temperature changes occur in regions where the *midHolocene* temperature signal itself is small (locations where the surface temperature changes are larger than 0.5°C are

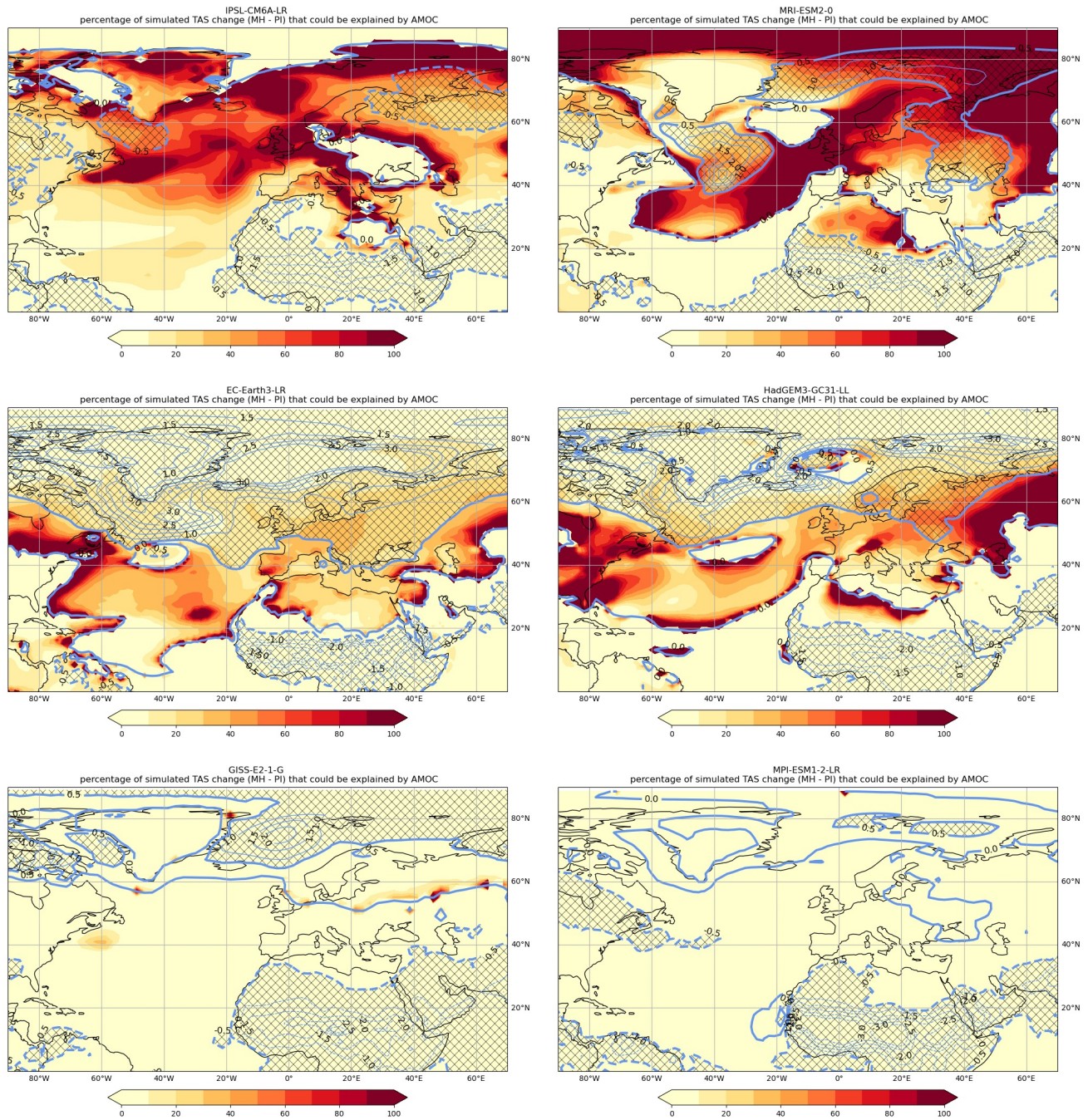

**Figure 5.** The maximum percentage of *midHolocene* simulated surface air temperature changes that could be explained by AMOC changes (*midHolocene - piControl*) for six different models. The top 4 panels show those models with a maximum AMOC strength change of 5% or more at 30°N. The bottom panels present two examples of models with minimal changes in overall AMOC strength. The overlaid contours show the magnitude of the *midHolocene* surface air temperature changes themselves (in °C). Negative changes are shown with dashed lines, positive changes have solid lines, and locations where the absolute size of the changes are larger than 0.5°C are hatched.

hatched in Fig. 5). For HadGEM3-GC31-LL and EC-Earth3-LR, this means that only regions out the northern North Atlantic are highlighted. Both IPSL-CM6A-LR and MRI-ESM2-0 show *midHolocene* temperature changes larger than 0.5°C in the subpolar gyre (notable for the present-day 'warming hole', Keil et al., 2020). Despite this, those regions demonstrate some of the weakest potential impact from *midHolocene* AMOC changes suggested across the North Atlantic and Nordic Seas for each model (Fig. 5). Unsurprisingly, neither MPI-ESM1-2-LR nor GISS-E2-1-G, which both have very little changes in AMOC strength in *midHolocene*(Tab. 1), suggest a minimal contribution to surface temperature changes from AMOC across the North Atlantic and Nordic Seas (Fig. 5). After analysing all the models, we conclude that the AMOC does not play a globally important role in explaining the temperature changes, and the role may even be secondary in the North Atlantic basin to other factors. This conclusion also applies to sea surface temperature, and holds for the *lig127k* as well (not shown).

A similar analysis can be performed to look at AMOC-related precipitation changes (Fig. 6), although here only the ensemble mean is presented rather than values for individual models (see Sect. 2 for methodology). The rainfall patterns associated with AMOC variations in both the *midHolocene* and *lig127k* experiments are similar. The majority of regions across the global show the AMOC contributing 10% or less of the precipitation changes (hatched in Fig. 6). Only a small portion of the Central Equatorial Pacific demonstrate at least 50% of the precipitation changes could potentially be explained by the AMOC changes (neither hatched nor stippled in Fig. 6). It is unclear whether there is physical reason for a strong AMOC influence in this particular region, and perhaps this is instead aliasing the damping of ENSO seen in the simulations (Brown et al., 2020). It further confirms our conclusion that, in practice, few precipitation changes can be explained by the AMOC changes. To summarise, the AMOC does not play a big role in explaining precipitation amount changes globally, and our analysis questions whether precipitation changes should be used as an AMOC proxy.

It is established that AMOC variations alter the location of the intertropical convergence zone (ITCZ), especially over the Atlantic (e.g. Braconnot et al., 2007; McGee et al., 2014). This is a feature whose strength is model dependent (Fig. 4), and depends interhemispheric heat transports and its feedbacks (Moreno-Chamarro et al., 2020; Buckley and Marshall, 2016). A characteristic dipole pattern associated with northward shift of the ITCZ emerges in the ensemble mean, even though its magnitude explains only a small portion of the interglacial changes in rainfall amount (Fig. 6). Further work is needed to quantify the ITCZ shift seen in the models, perhaps using the metric of Braconnot et al. (2007). However, previous work suggests the shifts in ITCZ are likely less than 1°latitude (McGee et al., 2014).

## 5 Discussion

Past changes in overall AMOC strength, especially its depth-integral, are difficult to reconstruct. Many previous studies have instead focused on examining individual components of the AMOC or inferred changes in deep water mass geometry (e.g Kissel et al., 2013; Solignac et al., 2004). However, one proxy technique is to use sedimentary Pa/Th (e.g. Yu et al., 1996; McManus et al., 2004), although modern geochemical observations highlight the contribution of other factors controlling the Pa and Th distribution (Hayes et al., 2013). For example, Missiaen et al. (2020) using a Pa/Th enabled model revealed that the

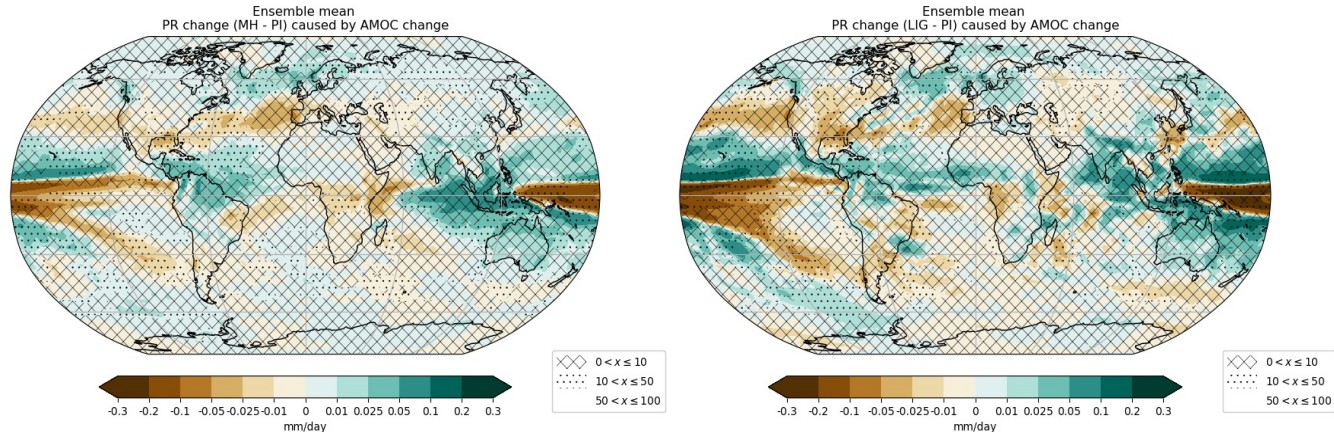

**Figure 6.** Ensemble mean plot of the precipitation changes caused by AMOC changes ($\overline{\Delta P_\Psi}$) at the *midHolocene* (left) and *lig127k* (right). Overlaid markers provide, the percentage of those changes that could potentially be explained by AMOC changes ($100 \times \overline{\Delta P_\Psi} / \overline{\Delta P}$): no shading indicates that AMOC contributes half or more of the changes ($>50\%$) seen in the experiment, whilst the dotted symbol indicates a small contribution (10-50%). Areas where there this no AMOC contribution ($<10\%$) are covered by crosshatching.

changes in biogenic particle fluxes can affect the Atlantic Pa/Th records, and the particle fluxes changes have been suggested
to cause far-field Pa/Th variations as well.

High-resolution Holocene Pa/Th reconstructions from the North Atlantic (Hoffmann et al., 2018; Lippold et al., 2019) show no observable changes, unlike the faint AMOC weakening in the Holocene shown by low-resolution Pa/Th data (Negre et al., 2010; Lippold et al., 2016; McManus et al., 2004; Gherardi et al., 2009; Ng et al., 2018). As the high-resolution Pa/Th records are both single-site studies from the subtropical Northwest Atlantic, it is unclear how well they represent the overall AMOC
strength (Hoffmann et al., 2018; Lippold et al., 2019). Taken together, the Pa/Th records indicate relatively similar AMOC strength for the mid-Holocene and preindustrial. There are fewer sedimentary Pa/Th records for the last interglacial, although they also do indicate substantial changes in AMOC strength (Guihou et al., 2010, 2011; Böhm et al., 2015; Jonkers et al., 2015)

Reconstruction of changes in the density profile of the Florida Straits show little changes in the strength of the upper limb of the AMOC over the past 8000 years (Lynch-Stieglitz et al., 2009). Just under half the Florida Strait flow is associated with
the AMOC, with the remainder relating to the wind-driven surface gyre circulation. Therefore the reported slight increase (4 Sv increase on a flow of 28-32 Sv) over the past 8000 years may be attributed instead to a strengthening of the wind-driven gyre circulation in the Western Atlantic (Lynch-Stieglitz et al., 2009). To our knowledge, an equivalent reconstruction does not exist for the last interglacial. In addition, the Bengtson et al. (2021) benthic $\delta^{13}C$ compilation shows no obvious changes in the spatial structure (latitudinal and depth extent) of the North Atlantic Deep Water (NADW) between the last interglacial and
mid-Holocene, and suggests that the mean NADW transport was similar.

In summary, no palaeo-reconstructions demonstrate substantive changes in the depth-integrated AMOC strength between either of the two interglacial states and the *piControl*. This, therefore, does not disagree with the PMIP4 ensemble demonstrating no consistent response in overall AMOC strength to the changes in orbital forcing (Sect. 3.2). However, it is not yet possible

to confidently assert that the PMIP4 ensemble is simulating the correct response. Two obstacles need to be overcome before that can happen: (i) a greater number of proxy records obtained throughout the basin, especially during the last interglacial, and (ii) uncertainties in the proxy data and their interpretation would need to be reduced significantly. Greater application of proxy system models (e.g. Burke et al., 2011) and proxy-enabled ocean general circulation models (e.g Sasaki et al., 2022; van Hulten et al., 2018), possibly combined with data assimilation approaches (e.g. Rempfer et al., 2017; Osman et al., 2021) could potentially resolve the latter.

Further research into the various flow components of AMOC and their respective coupling to the climate system is required, before one could conclude that there were no significant interglacial changes in AMOC. It is also worth noting that all the simulations and analysis here is looking at equilibrated timeslice simulations, rather than transient simulations (e.g Bader et al., 2020; Braconnot et al., 2019). Our conclusion of a minimal role for overall AMOC strength changes does not, therefore, apply to abrupt events where an AMOC response has long been identified (LeGrande and Schmidt, 2008).

## 6 Conclusions

The changes in mean AMOC strength in the *midHolocene* and *lig127k* have been investigated in this study using the PMIP4 models that performed the *midHolocene* and *lig127k* experiments, and they have been compared to the AMOC strength in the *piControl* experiment, respectively. Meanwhile, comparisons between the mean state of AMOC in the *midHolocene* and the *lig127k* have been made based on the ensemble mean. We further looked at the coherency across the two past interglacials for the forced response in AMOC, as well as the strength of the signal. A series of tests have been devised and four criteria identified to confirm an orbitally-forced response.

In all, the overall AMOC strength between either *lig127k* or *midHolocene* and *piControl* has not markedly changed in individual or model-averaged simulations (Fig. 1, Fig. 2). The two models that show the largest changes in the *lig127k* experiment change in the opposite direction. Many of the models show changes of amplitudes that could be explained by internal variability, rather than an forced response (Williams et al., 2020). It therefore seems the changes in orbital forcing in both the *lig127k* and *midHolocene* experiment have very limited impact on the overall AMOC strength. This finding is not inconsistent with available proxy reconstructions. Obvious differences in the AMOC strength between individual models reveal that the climate models are still struggling to accurately simulate the strength of the AMOC, as well as to capture the depth profile of the AMOC (Eyring et al., 2021).

After investigating the changes in AMOC during the interglacials, we explored the AMOC roles in the surface climate. The spatial patterns arising from internal variability in the AMOC remains largely unchanged between the *midHolocene*, *lig127k* and *piControl*, although there are variations amongst the models in those patterns (Fig. 4). We demonstrate that the AMOC does not play a globally important role in explaining interglacial temperature changes in the majority of the PMIP4 models (Fig. 5). Similarly, AMOC contributions to precipitation changes during the *midHolocene* and *lig127k* occur in very few regions across the globe (Fig. 6), with the sole exception being the Northern Equatorial Pacific Ocean. Therefore, we recommend caution when interpreting hydrology-related proxy reconstructions as providing information about the AMOC, especially if

**Table A1.** Digital Object Identifier (doi) for each simulation from CMIP6. The web address can be created manually by adding https://dx.doi.org/10.22033/ESGF/ in front of each doi. N/A in the Table indicates either that the simulation has not been performed, or that streamfunction data has not been uploaded to the Earth System Grid Federation.

| Model | Reference | *midHolocene* | *lig127k* | *piControl* |
|---|---|---|---|---|
| CESM2 | Otto-Bliesner et al. (2020) | CMIP6.7674 | CMIP6.7673 | CMIP6.7773 |
| EC-Earth3-LR | Zhang et al. (2021) | CMIP6.4847 | CMIP6.4798 | CMIP6.4801 |
| FGOALS-f3-L | Zheng et al. (2020) | CMIP6.12014 | CMIP6.12013 | CMIP6.3447 |
| FGOALS-g3 | Zheng et al. (2020) | CMIP6.3409 | CMIP6.3407 | CMIP6.3448 |
| GISS-E2-1-G | Kelley et al. (2020) | CMIP6.7225 | CMIP6.7223 | CMIP6.7380 |
| HadGEM3-GC31-LL | Williams et al. (2020) | CMIP6.12129 | CMIP6.12128 | CMIP6.6294 |
| IPSL-CM6A-LR | Lurton et al. (2020) | CMIP6.5229 | CMIP6.5228 | CMIP6.5251 |
| NorESM2-LM | Seland et al. (2020) | CMIP6.8079 | CMIP6.8078 | CMIP6.8217 |
| INM-CM4-8 | Volodin et al. (2018) | CMIP6.5077 | CMIP6.5076 | N/A |
| MPI-ESM1-2-LR | Scussolini et al. (2019) | CMIP6.6644 | N/A | CMIP6.6675 |
| MRI-ESM2-0 | Yukimoto et al. (2019) | CMIP6.6860 | N/A | CMIP6.6900 |
| ACCESS-ESM1-5 | Yeung et al. (2021) | N/A | CMIP6.13703 | CMIP6.4312 |

they pertain to rainfall amount rather than location. Combined with the inconsistent simulated forced response of AMOC during the PMIP4 timeslice simulations, the fingerprint analysis suggests that the overall AMOC strength changes should only be invoked to explain climate changes during abrupt events in interglacials.

**Appendix A: ESGF Digital Object Identifier (doi)**

*Code and data availability.* Monthly output from each simulation can downloaded from the dois listed in Table A1. The code used for plotting the figures in this manuscript and all the processed output fields are available at the Github repository: https://github.com/pmip4/AMOC-during-the-interglacials-in-PMIP4-simulations-.

*Author contributions.* ZJ performed the bulk of analysis and writing. CB and DT conceived of the project and supervised ZJ during the research. S.S. contributed to the text related to the last interglacial. CB modified and deployed the Climate Variabiltiy Diagnostics Package, as well as editing the text.

*Competing interests.* The authors declare that they have no conflict of interest.

*Acknowledgements.* We would like to thank all the modelling groups who performed the PMIP experiments and generously made the simulations output freely available. We also thank Zarina Hewett as an internal reviewer for her comments on this manuscript, and Adam Phillips and Jon Fasullo for developing the CVDP.

.

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
