# Peer review of "No changes in overall AMOC strength in interglacial PMIP4 timeslices"

_Climate of the Past, 2022_

## Author Response (AR3)

**Reply to the reviewers' comments: cp-2022-63**

Z. Jiang et al.

Correpondence: z.jiang.17@ucl.ac.uk

We would like to thank the 2 reviewers and the editor for their comments on our manuscript. We adopted most of the remarks.

Blue text below is our response to the reviewer's comments (reproduced in black).

**Reviewer's Comments #1:**

Line 7: better introduce the examined time periods in the abstract instead of using experiment abbreviations such as lig127k.

We have reworded the abstract to introduce the examined time periods and to remove the experiment acronyms.

Line 33: its not so clear to me how past changes in the AMOC can "... explain the recent global temperature changes..."

Sorry, it is our mistake that the references for '... explain the recent global temperature changes...' are talking about recent AMOC and temperature, changes etc. They do not support 'how the past AMOC changes can explain recent temperature changes'. This sentence has been removed in the revised manuscript.

Line 38: during tWo interglacials

**Corrected from 'tWo' to 'two'.**

Line 38-39: midHolocene is the simulation's abbreviation, not an interglacial. The abbreviation lig127k should be defined (last interglacial is not mentioned before). Further, 6 and 127 ka are time slices no period. What is the rationale for choosing this time slices. With >10 kyrs of duration for the Holocene, what makes 6 ka so special? Why not e.g. 4.2 ka representing the onset of the last Holocene subdivision? Same for the Eemian.

We added further introduction the two interglacials -- the Holocene (11.5ka BP - 1950 CE) and Last Interglacial (130-115 ka BP) in the paragraph. We have inserted a definition for the abbreviation of 'lig127k' in the paragraph. We have also provided some rationale for choosing the time slices -- they are selected by the PMIP, have been added in the revised manuscript.

Thank you for reminding us that mid-Holocene (the period of time) and *midHolocene* (the experiment) are not interchangeable. We have gone the rest of the revised manuscript checking for other instances of such imprecision.

Line 45: GHG is not introduced.

A definition for the abbreviation 'GHG' has been added.

Line 46: "...interglacial period with orbital forcing being the main difference" to what? "..., while other forcing" which?

Greater precision about the differences between the 2 timeslices, especially about the change in non-orbital forcings has been added.

"...remains very similar to thepiControl..." add space.

Space added.

Line 61: what would be inappropriate methods?

We meant "necessary" rather "appropriate" methods. We have changed the clause to read "After introducing the methods used in this study" instead.

Line 76: what is meant by "data availability". All available models have been used? Are there others? Why are these not available?

A model must have performed an experiment following either the protocol for the MH or lig127k as laid out by Otto-Bliesner et al., (2017), and then archived the output of this experiment onto the Earth System Grid Federation. We have additional sourced data from HadGEM3-GC3.1-LL, for this particular field is not available on the ESGF. We have rewritten this sentence to clarify.

Line 77: 'msftmz' and 'msftmyz' are cryptic and non-introduced abbreviations not understandable by non-specialists.

We feel that it is important to provide the specific fields used. They are non-sensical abbreviations, so we haven't specified it. Rather, we have rephrased the sentence to hopefully make it clear that these are technical terms, whose meaning does not need to be understood by all readers.

Line 79: "The data from FGOALS-f3-L ..." this is a very detailed information about a single model output and should be moved from the main text.

We have removed this information. We have added a short note about this into the caption of Table 1.

Line 87: "... is computed as zonal average..." what does that exactly mean. From the very eastern to western grids coast to coast? Is there a water depth threshold?

We have removed this sentence to avoid confusion, in part as it should have read zonal integral rather than zonal average. The computation of the meridional overturning streamfunction requires the conservation of mass in the meridional plane. So, all water movement is considered anywhere in the basin to satisfy this requirement.

Tab.1 : what is the significance of the Length parameter? It differs up to  $\sim$ 25 times between the simulations.

The run length is determined by each modelling group, and we have no control over it. When the mode reaches its equilibrium state for the chosen timeslice, the model will keep running for another several hundred years (the run length). It actually differs up to 12 times, as the very low run length comes from FGOALS-f3-L model, where we use historical experiment (1850-1899) in this model to substitute the PI experiment due to data availability. We feel this is important information to provide to the reader to allow them to assess our work, hence our inclusion of it in Tab. 1.

I'm concerned about the huge range of resulting AMOCs for PI (12 to 33 Sv). What are the (likely) reasons for some models clearly under/over- estimating PI AMOC? I think this an important lesson we can learn from this study.

We agree that the spread of AMOC across the model ensemble is concerning. This has been a major motivator for the discipline for over a decade, and is discussed in the recent IPCC report. We do not feel that this article is the place for further analysis of the preindustrial simulations. Nonetheless we have added further discussion of the issue into the manuscript in Sect. 3.1.

We feel that the main difficulties occur due to the following reasons:

(i) Some low resolution models lack overflow parameterization, and they cannot resolve some small scale features, such as the eddies and upwelling systems (Danabasoglu et al, 2014).

(ii) The intensity of the deep convection simulated in the Irminger, Labrador and Nordic Seas is sometimes too strong, and the wintertime convection can be overestimated in some models (Huang et al, 2014).

(iii) Models do not simulate the Iceland-Scotland Overflow Water (ISOW) flow path realistically, hence, the influence of ISOW changes on AMOC changes were underestimated (Lohmann et al, 2014).

(iv) The depth profile of the RAPID observations attains maximum values at 1,000 m depth. Models which particularly fail to capture the depth profile of the AMOC as observed by the RAPID array attaining maximum values at a shallower depth (700 - 800 m) than observations. Despite Boucher et al's (2020) suggestion for the IPSL model, the vertical profile is still realistic, just the magnitude is lower, the simulated depth profile still differs from the RAPID array observations.

(v) The simulated AMOC strength from the PMIP4 models use depth coordinates across all the latitudes, while in observations, we use density coordinates for high latitudes (OSNAP,

Lozier et al 2019), as it captures the physical processes more realistically due to the horizontal rather than vertical (occurs in subtropics) separation of the northward and southward limbs in the subpolar regions. When the streamfunction in density coordinates is remapped into depth space (Foukal et al, 2022), it allows direct comparison with the model simulated results. The remapped result shows the maximum AMOC occurs at subpolar rather than at subtropics. This is due to the inability for the depth coordinates to capture the horizontal circulation at subpolar, which lead to the maximum streamfunction in the Labrador Sea Water (LSW) Cell shifts southward into subtropics in the depth coordinates. Additionally, an underestimated strength of the Greenland-Iceland-Norwegian (GIN) Cell (70° N) is revealed in the depth coordinates. However, the inaccurate representative of the water mass transformation processes cannot be resolved easily, as the modelling groups need to re-assess the density and velocity fields, and re-calculating is time-consuming and may require additional computational resources.

(vi) The simplification of complexity of the overturning process in some models, and they may not have well-presented convection processes or convection-downwelling exchange processes (Kusters et al 2020), which are due to the limited computational resources.

Instead of showing the individual simulation compared to themselves in Fig. 1, it would be more helpful to show an inter-model comparison better reflecting the absolute uncertainties on the resulting AMOCs, as well as highlighting unrealistic outputs. Line 120: I can't see the feature described in front of "Fig.1" in the figure. Fig.1: the legend and caption of Fig.1 is very confusing. Actually whole Fig. 1 is confusing. E.g. what is the orange triangle? What are the Brierley et al 2020 simulations, and why are they shown at all? The numbers of symbols is higher than the simulations listed in Tab.1. Why are there no error ranges shown for individual simulations? Maybe indicate OSNAP and RAPID by arrow and name. Sometimes 30° and 50° is indicated sometimes not. Please revise the concept of the figure.

We agree. We were originally motivated to replot a *lig127k* version of Fig. 10 from Brierley et al (2020). We completely revised the structure of Fig. 1 to highlight the AMOC changes (rather than their piControl strengths.

In particular, when (from line 96 on) secondary fingerprints of AMOC on the wider climate patterns are examined. The relevance of modelled T and precipitation is questionable, when the underlying AMOCs are already inconsistent (e.g. in line 216 the authors claim that there is a strong relationship between AMOC changes and T).

Computing the AMOC fingerprint on the temperature and precipitation was motivated by previous publications discussing proxy records, as they can assume there are links between their local temperature/precipitation reconstructions and AMOC changes. In other words, sometimes remote reconstructions are interpreted as suggesting changes in AMOC during the Holocene. This analysis hopes to convince researchers not to interpret records in such a way. We have altered the introduction to section 4.2 to allude to this.

Line 130: "a changes"

**Correction made.**

Line 141: check grammar

**Done.**

Line 156: what is meant by shallow and deep branches. Its not clear to me what the paragraph is telling us.

Shallow branches refer to top 1000 m at low-mid latitudes, and deep branches refer to 0-60°N below 2000 m. The description in this paragraph was aimed at helping readers to grasp the relevance of the changes discussed in the following paragraph. It has been reworded to make this more obvious.

Figure 3: its hard to see what we can learn from this figure with the x-axes varying for each simulation. Please keep the axis constant to allow for inter-model comparison.

We have re-plotted the Fig. 3 to make the x-axis constant across all models. Thank you for spotting this.

Line 181: grammar

**Corrected**

Line 204: I don't understand this sentence.

The main purpose of this sentence is to explain why the numbers we provide for CESM2 are not identical to those included in Otto-Bliesner et al (2021) describing the CESM2 simulations. The value given in Otto-Bliesner et al (2021) is greater than the 1 Sv threshold, until the values we use in this work. This occurs because of slightly different definitions used to define the AMOC strength – ours is at a fixed latitude, whilst theirs is the maximum across all latitudes. We have edited the sentence to hopefully make it easier to understand.

Line 208: show(s)

**Correction made.**

Line 235: it's not clear to me why the similarity to hosing experiments demonstrates any appropriateness.

We've added a second sentence to explain the relevance of it. Fundamentally our analysis assumes that the fingerprint arising from internal variability (i.e. within a run) is the same as that of a forced response (i.e. between runs). This may not be the case, but looking at hosing (forced) experiments suggest that the differences are not going to be substantial.

Section 4.2: this part seizes a considerable part of the manuscript. In my opinion, this section should be shortened down and the study should keep the focus on AMOC itself, as the reader would expect from the title.

We have tried to make this section more concise. The other reviewer suggested some additions to the sections, so it may not end up shorter in the revised manuscript. In response to the reviewers earlier comment about the fingerprints, we have also added a short motivation.

Figs. 4+5: what are the criteria for showing "selected" model outputs.

We chose models which show more obvious AMOC's fingerprint on temperature and precipitation (Fig 4) and higher AMOC contributions to temperature (Fig 5) - showing too many figures with not-significant plots seemed excessive. We have not explicitly mentioned this motivation in the revised manuscript.

Fig.5 caption: what is "low contribution"?

We have added an explanation of it in the revised caption. It refers to the AMOC could only explain <10% of the temperature changes for the entire North Atlantic and Nordic Seas. Note that we have also altered this figure in light of the editor's suggestion.

Fig. 6: due to the hatched areas there is little to see, please revise concept of figure.

We feel that revising the figure to remove the hatching would be detrimental. Our conclusion from the section of the analysis is that the AMOC plays only a little role in the precipitation changes seen across the globe in the *midHolocene*. We feel this is well-represent visually by the hatching covering up the colours.

During our consideration of this comment, we discovered that our colour scale had disguised any regions of drying. This has been corrected in the revised figure, and we have expanded the key.

Section 5: compared to section 4 the discussion is too short and only comprises the comparison to observational data. Better insert here a detailed discussion on why the models partly differ so strongly from OSNAP, RAPID and from each other. Besides, it is interesting to see that paleoceanographic reconstructions by PA/Th indicate similarly constant interglacial AMOCs.

We have expanded this section to provide more detail about the paleoceanographic reconstructions. We have not discussed the ability of the models to capture present-day AMOC further at this point. We believe that our additions on the topic in Sect 3.1 are sufficient.

Line 276: Bradtmiller et al. 2014 focuses on Heinrich AMOC. Better cite the ground-breaking McManus et al. 2004 along with Yu et al. 1996 when it comes to introducing the proxy.

**Citation changed from Bradtmiller et al., (2014) to McManus et al., (2004).**

Line 279: There are a number of (low resolution) Pa/Th time series available, which, however, comprise more data points for the Holocene than the ones mentioned here: Negre, C., et al. (2010) (Nature, 468); Lippold, J., et al. (2016) (Earth and Planetary Science Letters, 445), showing indeed a faint AMOC weakening during the course of the Holocene. This is, however, NOT supported by both high-resolution studies, which indicate no observable change but stay fairly constant.

**That is a good point and we have rewritten these sentences accordingly.**

Line 282: Guihou et al. 2010 is not sufficient for refering to the existing data base. There are Böhm, E. et al. (2015). Nature, 517, Guihou, A., et al. (2011). Quaternary Science Reviews, 30, and Jonkers, L., et al. (2015). Earth and Planetary Science Letters, 419 covering this time period in higher temporal resolution, all indeed showing values in support of the statement made at line 283.

We now mention these studies. Thank you for making us aware of them.

Line 293: In the meantime there are a number of more sophisticated recent proxy-enabled models available giving more information on proxy related response uncertainties: Missiaen, L., et al. (2020). Climate of the Past, 16; Rempfer, J., et al. (2017). Earth and Planetary Science Letters, 468; Sasaki, Y., et al. (2022). Geosci. Model Dev., 15; van Hulten, M., et al. (2018). Geoscientific Model Development, 11.

We have now expanded on the sentence that introduced the future work needed to confirm the absence AMOC changes in the models are occurring appropriately. This expansion mentions the promise of proxy-enabled models as a necessary step (using the citations recommended).

Line 285: I don't understand this sentence.

We have rephrased the sentence about the Florida Strait. Hopefully, this make is clearer that AMOC is only one of the factors determining its transport.

Line 314: This statement is very true and I expected more detailed explanations for this inconsistency.

Many existing publications have been devoted to understanding the differences in preindustrial AMOC strengths in CMIP6, and we feel that this article should not be the vehicle for further analysis of them. In response to earlier comments, we have increased the discussion of this in Sect 3.1.

Nonetheless, it was an oversight to have not mentioned this again in the conclusion. We have now introduce an sentence, pointing readers to the IPCC assessment of this issue.

**Reviewer's Comments #2:**

L45 The continental configuration should be specified. It will also be useful to include a figure showing the insolation differences between interglacials and PI as a function of calendar month and latitude.

We have now added a note about the continental configuration to this paragraph. We appreciate the importance of the changes in insolation and see where the reviewer is coming from. However, we don't feel that it necessary for a figure to be included in manuscript. Instead we have upgraded the citation to Otto-Bliesner et al. (2017) to explicitly provide the relevant figure as well (Fig. 3b).

L39&L78&Table1 More models doing MH simulations compared to LIG simulations should be consistent with the definition of Entry card.

You are, of course, correct. However, given that the number of simulations is so similar it seems unnecessary to state this explicitly.

L87&L91 The observation periods of RAPID and OSNAP are just a couple of years, and should be reflected in the comparison and discussion. The difference in AMOC strength between PI and present-day can be mentioned (e.g. Dima et al. 2022).

The observation periods of RAPID (since 2004) and OSNAP (since 2014) have been added in Sect. 2.

We have mentioned that the trend over the historical period cannot be an explanation of the differences between the models and observations as they are too small.

L104-110 Please reorganize this part of data processing to make it easier to read.

We have rewritten the sentences. We hope that this does indeed make it easier to read.

L143 Max AMOC strength at 10.3 Sv is very small.

We added discussions on the possible reasons for the underestimated IPSL PI AMOC in Sect. 3 - in response to Reviewer 1's comments.

L187 onwards I think a spectral analysis can be more accurate in estimating low frequency variability of AMOC. Also, the different lengths of each simulation can affect the significance of r2.

We had already performed the power spectra analysis on each model. However, some models have relatively short simulation years, it is difficult to visualise, especially when trying to figure out the lowest frequency in the simulation by looking at the plot. Therefore, we prefer using the r2 to assess whether the model has substantial low frequency variability.

Meanwhile, we have computed the p-value of the correlation coefficient. The r-value of all models in all 3 experiments are statistically significant (p<0.05), with only FGOALS-f3 PI experiment has a r value not statistically significant, possibly due to the short run length of just 50 yrs, as we use historical experiment in this model to substitute the PI experiment (see Tab 1 captions). This is now mentioned in the caption of Tab. 2

L205 Please rephrase 1.3x here.

We have rephrased this sentence to refer to the signal ratio - like in Tab. 2

Sec 4.2 The impacts of AMOC on the N-S surface temperature gradient and shift in precip center (e.g. ITCZ location) can also be quantified by regression. The conclusion in L272 may need to be modified.

We agree that it could in principle be quantified. However, Reviewer 1 has commented that Sect. 4.2 already seems overly long. Instead of including analysis, we have mentioned the potential shift of the ITCZ through citations to the literature. The correction to the colorscale in Fig. 6 (see earlier) makes a dipole pattern in emerge in the Tropical Atlantic that coincides nicely with this story. We have also made it explicit that our findings only hold for local precipitation amount.

L322 This is a strong statement. I feel it is not fully supported by the current results. More analysis on the impacts of AMOC on precip center (as I mentioned before) can make it more convincing.

We have reduced the strength of this statement – in part by stating that it only holds for local precipitation amounts, rather than shifts in patterns.

L279 The reconstructed AMOC provides useful constraints for the model. These results should also be included in the abstract.

Thank you for this suggestion. We have now added such a sentence to the abstract.

L287 Provide more details for changes in the atmospheric circulation.

We have now rephrased this sentence – in light of Reviewer 1's query about it. So we feel this is not now necessary

L298-300 In an equilibrated state there can be intrinsic multi-centennial AMOC variability, e.g. as proposed by Li and Yang (2022), even after the abrupt events.

Yes, it is correct. We have noticed that the multi-centennial AMOC variability does exist in the TraCE-21k transient simulation too. However, we are not sure this point is relevant to this particular manuscript, so have not included it in the revised manuscript.

Fig. 4 The surface temperature changes due to AMOC forcing are robust in the Nordic Seas. I am curious how the sea-ice would respond and potentially feedback to AMOC strength.

Unfortunately, we feel that the sea-ice feedback must be left for future work, as our current regression-driven approach is not sufficient to identify the direction of any feedbacks.

**Reply to the Editor's comments: cp-2022-63**

Z. Jiang et al.

Correpondence: z.jiang.17@ucl.ac.uk

We would like to thank the editor for the remarks about our response to the two reviewers' comments and the remaining errors in our manuscript. We adopted most of the remarks.

**Blue text below is our response to the editor's comments (reproduced in black).**

Table 1: It would seem appropriate to me to include references to the model descriptions and/or mid-Holocene and lig127k simulations. Readers might want to go back to the original work to obtain more information on methods and climatic states.

**Yes, perhaps we should not have left this information for the appendix only. We have now added a 'References' column to Tab .1.**

Please be more precise in the caption regarding "Simulation length". You are not showing the total length of the simulations, but instead are showing the length you are using for your analysis (I guess). This is probably determined by what is currently available on the ESGF website.

**You are correct in guessing that it refers to the run length after the spin-up procedure in each experiment. We have amended the caption accordingly.**

L. 179-180: There is a problem with that sentence.

**We have rewritten the sentence to make it short and less verbose.**

Figure 5 (and associated section 4.2): As is, this figure worries me and in fact could be misleading. For example, it shows that large areas of the Arctic/North Atlantic eastern Europe TAS changes are only due to AMOC changes (100%). Without seeing the absolute TAS changes between the mid-Holocene and PI, one could wonder whether the results shown in Fig. 5 are all relevant/significant. If the mid-Holocene TAS changes in a particular location are not significant to begin with, this percentage analysis might give biased results. In addition, deconvolving changes due to orbital parameters and AMOC might not be that straightforward. This can be illustrated by the results shown for the IPSL. It is noted that in this mid-Holocene simulation more than 40% of the changes can be explained by AMOC changes, while the AMOC is only 0.5 Sv lower, which is not statistically significant (as per Tables 1 and 2). So how come a non-statistically significant and very small AMOC weakening explain large areas of the mid-Holocene TAS changes? I suppose that one way around that would be to show the absolute mid-Holocene TAS changes for the simulations, with statistical significance overlaid and next to it show the absolute TAS changes that could be attributed to the AMOC, also including statistical significance. Please also modify section 4.2 accordingly.

There are two separate responses to this point. Firstly, we have changed the related text to introduce a necessary scepticism (for example altering 'can' to 'could'). Our approach only constrains that maximum possible from AMOC-related changes and ignores all other drivers (of which there may be many).

Secondly, we have modified Fig 5. The simulated 'absolute TAS changes' are overlaid onto the plot of 'the percentage of simulated TAS changes that could be attributed to the AMOC changes'. Meanwhile, we also cross-hatched the regions where the TAS changes are  $> 0.5^{\circ}$ C.

Now the plots clearly show the regions where the TAS changes are large (>0.5°C, crosshatched) are not the same regions as where the AMOC has high contribution (>50%) to the temperature changes seen in the midHolocene (clearly shown in the EC-Earth3-LR and HadGEM3-GC31-LL models). In other words, despite the region in the low-mid latitude North Atlantic shows ~40% of the TAS changes could be explained by the AMOC changes, the actual absolute TAS change is very small (<0.5°C). Among all the 11 PMIP4 models, only the IPSL-CM6A-LR and MRI-ESM2-0 models have small areas in the northern North Atlantic and Nordic Seas where both the TAS changes is great than 0.5°C and the AMOC could explain half or more (>50%) of the simulated TAS changes. In summary, AMOC does not play a big role in explaining the surface temperature changes in the midHolocene.

L. 251: The IPSL mid-Holocene AMOC is slightly weaker than at PI (and not statistically significant).

**Thanks, we have now altered the relevant sentences in the paragraph.**

Discussion: I agree with the Reviewers that a more extended discussion of AMOC inferences from paleoproxy records is needed. You might also want to look at https://cp.copernicus.org/articles/17/507/2021/, which suggested that there was no significant difference in the latitudinal and depth extent, as well as mean transport of North Atlantic Deep Water (NADW) between the LIG and the Holocene (7-2ka) using an updated d13C compilation.

In the revised manuscript, we have added further detail and discussion on the palaeo-data side using the existing literature of more detailed modern oceanography studies. This includes a short discussion of Bengtson et al (2021).

Dear Dr Menviel,

Thank you for you time looking over our revised manuscript. We accept that our original revisions do not fully address the reviewers' and your comments. We hope that you will agree with us that the new version of the manuscript does resolve them.

In summary, this version has substantially reduced the content in Section 4. This has involved removing the bottom two paddles of figure 5 and only providing figure 6 as supplementary information. This means that Section 4 now has no numbered subsections. We've also gone through and made a collection of edit to improve the readability of the piece, especially the description of Fig. 1. These edits all of the minor issues that you kindly pointed out, amongst several others.

Yours,

Zhiyi Jiang and Chris Brierley.